# Separability of Human Motor Memories during reaching adaptation with force cues

**Frédéric Crevecoeur**[1,2]*, **James Mathew**[1,2], **Philippe Lefèvre**[1,2]

**1** Institute of Information Technologies, Electronics and Applied Mathematics (ICTEAM), UCLouvain, Louvain-la-Neuve, Belgium, **2** Institute of Neuroscience (IoNS), UCLouvain, Louvain-la-Neuve, Belgium

* frederic.crevecoeur@uclouvain.be

**Data Availability Statement:** The data was deposited in DRYAD (https://doi.org/10.5061/dryad.b2rbnzsk3) and the code used to produce the simulations shown in Fig 6 was uploaded to

## Abstract

Judging by the breadth of our motor repertoire during daily activities, it is clear that learning different tasks is a hallmark of the human motor system. However, for reaching adaptation to different force fields, the conditions under which this is possible in laboratory settings have remained a challenging question. Previous work has shown that independent movement representations or goals enabled dual adaptation. Considering the importance of force feedback during limb control, here we hypothesised that independent cues delivered by means of background loads could support simultaneous adaptation to various velocity-dependent force fields, for identical kinematic plan and movement goal. We demonstrate in a series of experiments that indeed healthy adults can adapt to opposite force fields, independently of the direction of the background force cue. However, when the cue and force field were in the same direction but differed by heir magnitude, the formation of different motor representations was still observed but the associated mechanism was subject to increased interference. Finally, we highlight that this paradigm allows dissociating trial-by-trial adaptation from online feedback adaptation, as these two mechanisms are associated with different time scales that can be identified reliably and reproduced in a computational model.

## Author summary

The conditions under which humans can adapt reaching movements to different force fields in parallel remain the subject of active debates. Mounting evidence highlights that contextual factors linked to movement planning are necessary to form different motor memories. Here we show that background forces indicative of the direction of a force field could play the role of contextual factors, and enable dual adaptation. However, we uncovered that when the cue and the force field were in the same direction but of different magnitudes, the motor memories were still distinct but subject to increased interference. We further show that different timescales of adaptation observed can be explained by a combination of offline and online adaptation in a model of reaching control.

ModelDB (access to the model working code: http://modeldb.yale.edu/267586).

**Funding:** FC and JM are supported were supported by grants from Belgian National Fund for Scientific Research (F.R.S.-FNRS, Belgium, grant numbers 1. C.033.18 and T.0048.19), and by the "Action de Recherches Concertée" coAction (UCLouvain). The funders had no role in study design, data collection and analysis, decision to publish, or preparation of the manuscript.

**Competing interests:** The authors have declared that no competing interests exist.

## Introduction

Motor adaptation is the neural mechanism by which humans and other animals can learn to produce motor commands in anticipation of contextual contingencies. The anticipatory nature of motor commands and their flexibility suggested that adaptation is supported by the formation of internal models in the brain [1,2]. Internal models have become a widespread computational framework, encompassing the expression of priors for perceptual decision-making, sensorimotor learning, and formation of control policies for skilled movements. In the context of human reaching control, internal models manifested behaviourally by the fact that motor commands depend on adaptive representations of movement dynamics, and the properties of the underlying neural mechanism have been the subject of a considerable research effort.

Towards characterizing the formation of internal models for reach control, recent work has investigated the conditions under which different motor tasks can be learned at the same time. A clear motivation is to generalise laboratory observations about motor adaptation to everyday tasks. Indeed, on daily bases we manipulate different tools and it is not uncommon to learn to practice different sports or music instruments in parallel. When applied to force field learning, this question has been approached either by exposing participants to different perturbations applied following different schedules, or by changing the planning condition.

Interestingly, when visual or haptic cues indicate the occurrence of a force field in one or another direction, the formation of different motor memories is absent or limited [3–6]. It is suggested that two major factors condition the formation of different motor memories. First, the training schedule [5] and the separation of training phases in time may help the formation of different motor memories [7], although interferences may still occur [8,9]. Second, studies reported that different contexts, evoked by target remapping [10], visual cues [11–13], vestibular cues [14], limb configuration [15], and prior or follow through movements [16,17], enabled the acquisition of different motor memories of multiple force fields. The common property of these different learning experiments was that the contextual cue was independent of the sensorimotor transformation, and allowed that the movements were associated to distinct goals [18].

To date, this approach has not considered the possibility that external forces mays serve as a cue to enable dual adaptation, for identical kinematic plan and movement goal. Here we hypothesized that limb afferent feedback about external loads could also produce different contexts and planning states. The rationale for formulating this hypothesis was based on a growing consensus over the following observations: on the one hand, it appears very clearly that both anticipatory (*feedforward*) and feedback control adapt in parallel [19–23]. On the other hand, studies on feedback responses to mechanical perturbations have concluded that they are quickly tuned to an estimate of externally applied loads [24,25], making this variable a necessary component of state-feedback control models [26]. More generally, models of goal-directed reaching have often considered acceleration and forces as state variables estimated in the brain and used for control [27–32]. Thus, if the nervous system uses estimates of external and self-generated forces, it is conceivable that force cues prior to movement enable simultaneous adaptation to different force fields.

Here we demonstrate in a series of experiments that indeed background forces contributed to parallel formation of different motor memories. However, an unexpected finding was that the underlying mechanisms was subject to increased interference when the force field and the force cue were in the same direction with different magnitudes, which sets constraints on the use of force cues for dual adaptation as well as for our understanding of the underlying neural mechanism. Finally, we demonstrate that the use of background forces to hasten motor

adaptation may be a powerful paradigm to dissociate anticipatory adjustments across trials and within-trial feedback adaptation components of reach control [33].

# Results

## Experiment 1

Participants grabbed the handle of a robotic arm and performed visually guided, forward reaching movements of 15cm in amplitude. Clockwise or counter-clockwise orthogonal velocity-dependent force fields were associated with leftward or rightward constant background forces in two different contexts, and were randomly applied across trials. The contexts were defined as the sign of the relationship between the background force and the force field: in the positive context both forces had the same sign, in the negative context these forces had opposite sign. Participants performed two series of three blocks in each context, each consisted of force field and catch trials randomly interspersed (Fig 1A–1C, see Methods). It must be noted that the design including both positive and negative contexts ensured that the association between the background force and the force field could be dissociated from the agonist activity.

We first extracted the initial angle measured at the crossing of a virtual position threshold and observed a clear decrease across trials in each context (threshold: Fig 1B, initial angle

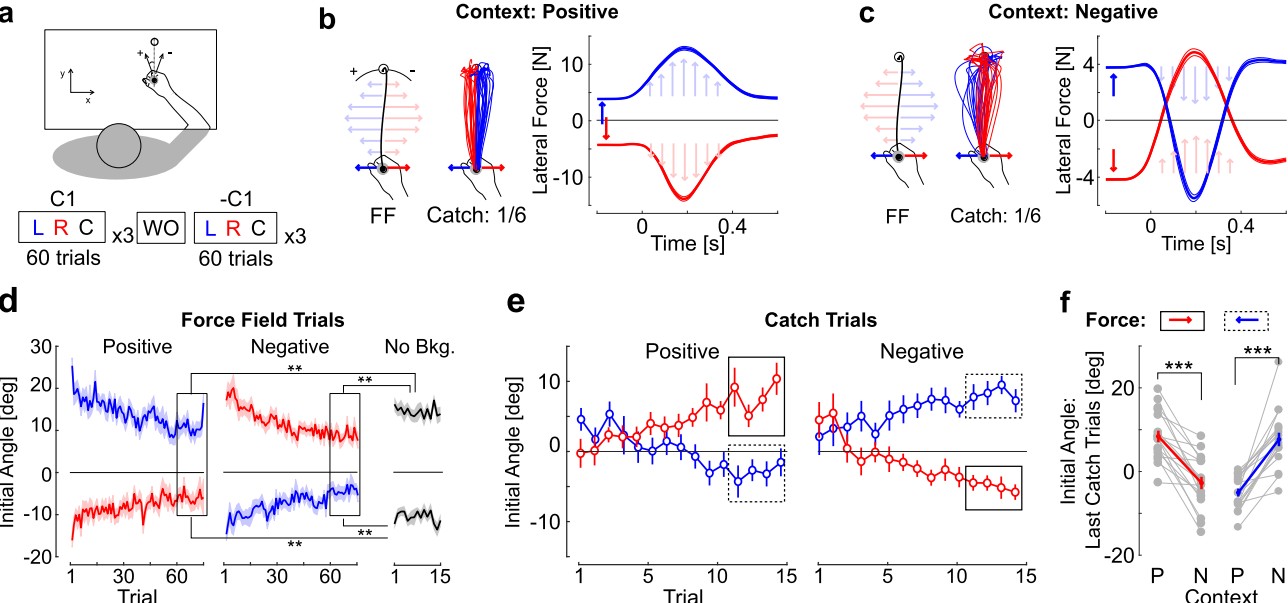

**Fig 1. Experiment 1. a.** Schematic top view of a participant performing the task and convention for the definition of the initial aiming angle (see Methods, positive angles correspond to CCW rotations); illustration of the time course of the experiment: participants performed 3 blocks in one context (C1) including Left and Right background forces with catch trials (L, R, C, respectively), followed by a washout block (30 trials) and another series of 3 blocks in the opposite context (-C1). **b.** (left) Relationship between background force and force field in the positive context (FF); (centre) Hand traces from each participant (n = 18) during catch trials featuring background force but no force field, observe that each trace is biased in the direction opposite to the background force; (right) Group average of force as a function of time in the positive context for force field trials. Thick and thin traces represent the grand average across participants and the standard error, respectively. Recall that force field and catch trials were applied randomly in each context. **c.** Same as panel b for the negative context, in which the force field is in the direction opposite to the background force. **d.** Initial angle as a function of trial number in each context (mean±SEM) and in the last block of force field trials randomly interleaved performed without background force (No Bkg., black). Two stars illustrate significant differences at the level $P<0.005$ across these trials and the last 15 trials from each context (insets). **e.** Initial angle during catch trials in each context (mean±SEM). Insets show the last four catch trials used for statistical comparison. **f.** Effect of the context on the aiming angle averaged across the last four catch trials in each context. Individual gray dots represent single participants' data, red and blue traces are the group average. For the same background force (red or blue trials), the context produced a clear and highly significant bias away from the corresponding force field, showing internal priors and anticipatory compensation for the force field (three stars: Paired t-test, $P<0.001$).

across trials: Fig 1D). In all cases, the position threshold used to calculate the initial angle was crossed before 150ms following reach onset (95th percentile of crossing times across participants was between 110ms and 145ms), that is, prior to any modulation in feedback responses during movement which was previously documented to occur in muscle EMG after 250ms in the same setup and task [34]. Thus, the measurement of initial angles was impacted by the same feedback responses in all cases.

It can be observed that the initial angle across trials with context-dependent background forces exhibited initial angles that were smaller in absolute values than the initial angles recorded when there was no initial background force (Fig 1D, No Bkg.). We calculated the mean initial angle across the last 15 trials in each context and for each force field, and compared it to the mean initial angle of across the 15 trials of the corresponding force field direction without any background loads. Every comparison reported highly significant differences: absolute initial angles for clockwise trials were smaller in the negative ($t_{17}$ = -4.72, $P < 10^{-4}$) and positive contexts ($t_{17}$ = -3.6, $P$ = 0.001); for counter-clockwise perturbations the initial angles were also smaller in the negative ($t_{17}$ = 4.03, $P < 10^{-4}$) and positive contexts ($t_{17}$ = -3.61, $P$ = 0.001, one-sided paired t-tests). This observation is compatible with the hypothesis that the background force enabled anticipatory compensation for the force field.

To establish that the compensation for the force field resulted from an adaptation of the controller instead of a default compensation strategy, we applied catch trials randomly in which the same background force was used, but the force field was unexpectedly turned off. The aiming angle during these catch trials displayed clear biases compatible with anticipation of the force field (Fig 1E). The average initial angle across the last four trials (Fig 1E, inset) was compared for the same background force across positive and negative contexts. There was a clear effect of the expected force field in each case, showing that participants partially compensated for the associated force field (Fig 1F: paired t-tests, rightward force, red, $|t_{17}| > 6.9$, $P < 10^{-5}$; leftward force, blue: $|t_{17}| > 7.9$, $P < 10^{-6}$, effect sizes > 1.3, power > 0.99).

There was a directional bias corresponding to the order in which participants experienced the positive and negative contexts, such that those who performed trials in the positive context first tended to display larger values of initial angles, however the comparisons of initial angles during catch trials performed on each subgroup revealed similar differences. For the leftward background force, the differences between initial angles during catch trials across contexts were 9.8±1.8deg (positive first) and 15.5±2.3deg (negative first, mean±SEM). For the rightward background forces, these differences were -10.5±2.5 (negative first), and -11.7±2 (positive first). We also observed that the initial angles at the end of the washout block did not completely return to values corresponding to baseline, although there was a clear tendency for a decay in absolute value. Observe that it does not impact our results because upon switching context, the carry-over from the first context was in fact biasing the initial angles in the wrong direction.

To summarize, participants exhibited initial angles during catch trials indicative of an internal anticipation of the associated force field. This association was flexible as they were able to relearn the novel, inverted mapping midway through the experiment upon changing of the context.

## Experiment 2

This experiment probed the ability of participants to learn about the magnitude of the background force indicative of the magnitude of the force field. Rightward background forces were always applied of 2.5N (light cue) or 5N (heavy cue), and an orthogonal velocity-dependent force field with proportional magnitude was applied (Fig 2A and 2B, light force field: $\theta$ =

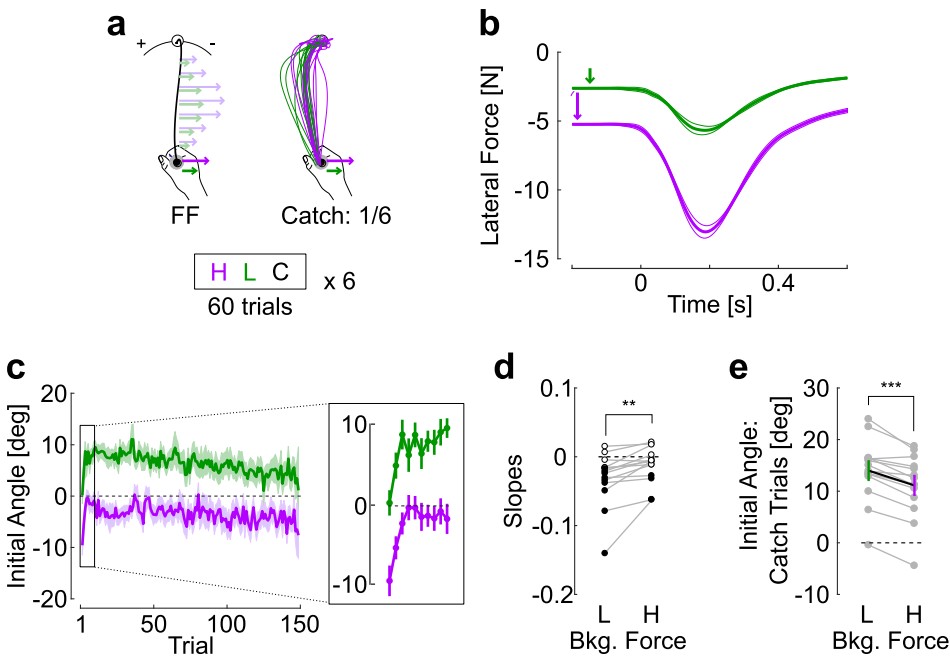

**Fig 2. Experiment 2. a.** Schematic representation of the task, participants (n = 14) performed reaching movements in a force field with same direction but difference magnitude: heavy background and force field in purple, light background and force field in green. Catch trials were applied with the same frequency as in Experiment 1. **b.** Time course of measured force as a function of time. The thick traces display the grand average across participants and thin lines delineate one SEM across them. **c.** Initial angle across force field trials for the different background and force field magnitudes. Traces are the mean across participants and shaded regions are the standard error of the mean. The inset magnifies rapid adaptation occurring over the course of the first 10 trials. **d.** Individual slopes during the second phase of the experiment. Filled and open dots designated linear regressions that were significant or not, respectively (level $P = 0.05$ for individuals' data). Paired comparison exhibited significance difference across light and heavy trials (two stars, paired t-test, $P < 0.005$) **e.** Initial angle for the catch trials, single gray dots represent individual participants and color traces are the population mean ($\pm$SEM, three stars, $P<0.001$).

7.5Nsm$^{-1}$, heavy force field: $\theta = 15$Nsm$^{-1}$). Light and heavy trials were randomly interleaved in blocks along with catch trials. We observed two clearly different learning phases corresponding to different time constants. The first was a very rapid zeroing of the initial angle for the heavy trials over the course of the first ~20 trials based on visual inspection (Fig 2C, inset). This strategy was expressed at the cost of overcompensating for the light force field which, as a consequence, displayed increasing values of initial angle.

Intuitively, it is possible that participants used a single internal representation for the two magnitudes. If the latter is tuned to the heavy trials, then the light ones become logically overcompensated. However, the second phase provided evidence that participants used distinct motor memories across light and heavy trials. We regressed the initial angle as a function of the trial index starting at indices $\geq 20$ for heavy and light trials (excluding the initial rapid phase). In addition to trial indices, the matrix of fixed predictors included an intercept, a categorical factor for the trial type (heavy or light), and a random intercept capturing inter-individual variability. This linear mixed model revealed a clear evolution across trials (linear mixed models, effect of index on initial angles: $|t|>10$, $P < 10^{-6}$), and a reliable interaction between the trial index and the trial type, thereby rejecting the hypothesis of a single representation for which no difference between loads, and thus no interaction term, was expected (interaction between load and index, $|t| = 3.9$, $P < 10^{-3}$). As a control analysis, the same linear mixed model was fitted to the forward hand velocity as dependent variable to verify that the

perturbations were comparable. This model revealed only a weakly significant effect of the trial index ($|t| = 2.05$, $P < 0.04$), and no significant interaction between trial indices and loading condition ($|t| = 0.95$, $P = 0.34$). Thus, the different evolution of initial angles across light and heavy trials was attributable to different anticipatory strategies associated with the background force. This tendency was captured for individual participants by fitting linear models to the same data (starting at trial 21, after the initial rapid phase). The individual slopes were clearly more negative and were significant in a majority of participants for light trials (Fig 2D and 2L), while the slopes for heavy trials were distributed around zero and only a few were significant. Paired comparison of the slopes across heavy and light trials were significant ($|t_{13}| = 3.5$, $P = 0.003$).

To corroborate the foregoing result, catch trials consisting of light and heavy background force cues but without force field also exhibited significant differences. Again, a single internal representation independent of the background force would produce the same initial angles. Instead, a strongly significant difference was observed across light and heavy catch trials (Fig 2E: paired t-test: $|t_{13}| = 6.59$, $P = 0.00017$, effect size: 0.46, power $> 0.99$). This result was unexpected. Indeed, larger initial angles associated with light force cues suggested that in absolute, the magnitude of the force field anticipated in this case was larger than the one associated with the heavy cue. We performed an additional analysis to verify whether any effect linked to early stages of movement were not missed. Indeed, it is conceivable that heavy catch trials produce larger initial errors, followed by larger stretch responses between reach onset and the time when the initial angle was measured. If we follow this rationale, in our case we would have observed an inversion with larger initial angle for heavy trials closer to reach onset. But this was not the case, suggesting that the light background force was indeed associated with a heavier force field. Experiment 3 further reproduced this result and highlighted the interference between motor memories cued with different magnitudes, in the sense that exposure to a trial type (heavy or light) impacted adaptation to the other type.

## Experiment 3

In this experiment, participants first performed a full block of trials with heavy loads only, prior to performing the remaining blocks identical to Experiment 2 with heavy and light trials interleaved. Thus, this experiment was similar to Experiment 2, except that the light trials were introduced after participants were exposed to the heavy trials. Interestingly, although values of initial angles were close to 0 after this first block, the introduction of the light trials randomly interleaved with heavy trials clearly produced an absolute increase in initial angle for heavy trials (Fig 3A–3C, compare B1 and B2 purple). In this case, it was the light force field that produced novel disturbances with larger errors, which were likely reduced at the cost of partial under-compensation for the heavy force field trials. The initial angles from heavy trials were averaged across the last 10 trials of the first block (Fig 3B, epoch B1) and were compared to the average of the first 10 heavy trials from the second block (epoch B2). This comparison revealed a large and highly consistent change in initial angle (Fig 3D, paired t-test: $|t_{14}| = 8.53$, $P < 10^{-6}$, effect size: 0.89, power $> 0.99$).

Thus, the introduction of the light force field produced interference in the sense that it impacted the recently acquired ability to compensate for the heavy one. Moreover, the inversion that we observed in Experiment 2 was reproduced, such that the catch trials associated with light background force had larger initial angle than the ones with heavy force field trials (Fig 4: paired t-test, $|t_{14}| = 3.57$, $P = 0.003$, effect size: 0.5, power = 0.84). Notably this effect was smaller than the effect reported in Fig 2D. This difference was quantified with mixed linear models in which the initial angle was fitted as dependent variable as a function of the trial type

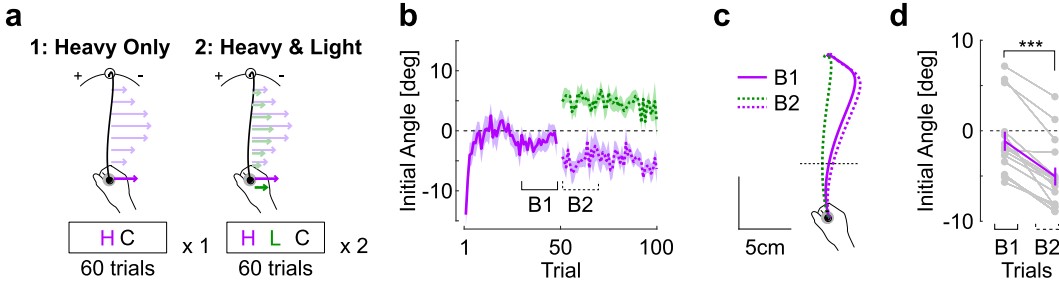

**Fig 3. Experiment 3. a.** Schematic representation of the task design, during which participants (n = 15) first adapted to the heavy trials (purple, large background and force field magnitude) for one block of trials prior to introducing the light background and force field trials (green) randomly interleaved with heavy trials in the second block. **b.** Initial angle across trials (mean±SEM) separated as follows: first block with heavy trials only (solid purple) followed by second and third blocks with light trials randomly interleaved. Trial indices refer to the series of heavy trials. The last 10 trials of the first block (B1) and the first 10 trials of the second block (B2) were used for comparisons. **c.** Grand average of hand traces prior (B1) and after (B2) the interruption between the first and second block. The introduction of light trials clearly deviated the initial angles of heavy trials. **d.** Initial angle averaged across B1 and B2 phases for the heavy trials. Gray dots represent individual participants' data; purple traces are mean±SEM in each group of trials (three stars: Paired t-test, P<0.001).

(light and heavy) and the experiment number as fixed predictors, an interaction between type and experiment, and a random intercept per participant (recall the two groups were distinct). The model revealed a significant interaction (|t| = 3.19, P = 0.0014), suggesting that the difference between initial angles across catch trials was reduced in Experiment 3. This result is compatible with a smaller interference between the two motor memories that possibly resulted from the adaptation schedule of this experiment, in which the light trials were introduced at the second block.

To summarise: Experiment 1 showed that a background force cue evoked partial adaptation to force fields of opposite direction randomly interspersed, and that that the underlying association could be formed in different contexts. Experiments 2 and 3 concentrated on the possibility that participants could also learn the magnitude of randomly interleaved force fields cued by background forces of different magnitude. We observed an interference, producing even an inversion in catch trials such that the light cue produced larger mirror-image perturbations than the heavy one. This result was reproduced across the two experiments. Besides this

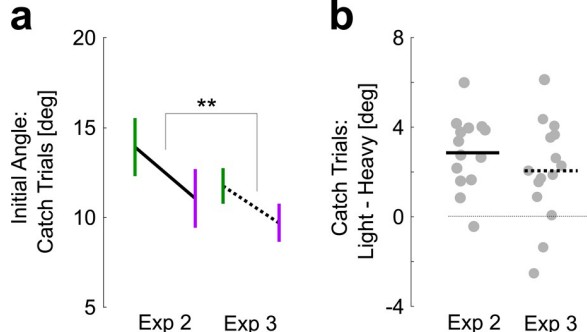

**Fig 4. Comparisons between Experiments 2 and 3. a.** Average initial angle during catch trials heavy (purple) and light (green) background forces. Solid data from Experiment 2, dashed from Experiment 3. Observe that heavy and light were under and over-estimated, respectively. The two stars aligned on the difference illustrate significant interaction between the difference between heavy and light, and the experiment from mixed linear models (see Methods, P<0.005). **b.** Difference between initial angles from light and heavy trials from Experiments 2 and 3. Dots are individuals' data; horizontal bars are group average.

interference, based on statistical modelling it was possible to show that distinct motor memories were formed by revealing an interaction between trial types and trial indices. The difference in initial angles during catch trials was depended on the schedule across Experiments 2 and 3. Although we found evidence for distinct motor memories associated with different magnitudes, these memories did not directly scaled with the force magnitudes and it remains unknown whether participants can learn to fully compensate for the two force fields after longer training.

## Movement adjustments across and within trials: Adaptive control model

We recently argued that trial-by-trial adaptation of movement control could result from the combination of online, adaptive state feedback control, and offline construction of motor memories [35]. The data of Experiment 1 offered a novel opportunity to illustrate this model, because online and offline mechanisms may exhibit different timescales revealed by the dual adaptation task. In a standard adaptation scenario, short term effects of both trial-by-trial adaptation and online adaptation are confounded, whereas in fully random adaptation tasks, only the online component can be extracted. Here, the dual adaptation scenario slows down the trial-by-trial component, while the random alternation of force fields likely recruited the online component at a faster timescale, thereby dissociating their effects and making them visible. We show that the model with adaptive state feedback control can accommodate this behavioural feature.

We concentrate on the grand-average of initial angles and path lengths across positive and negative contexts from Experiment 1. By convention we refer to trial-by-trial updates in the controller as the *"offline"* adaptation mechanism. We did not explicitly model the interference observed in Experiments 2 and 3 although different timescales could also be observed. In a standard trial-by-trial adaptation scenario, each trial is performed with a fixed controller corresponding to the current estimate of the force field parameter. This estimate evolves from trial-to-trial following offline processing of movement errors (Fig 5B, offline). In the adaptive control model, it is possible that the estimation about the force field parameter evolves online, leading to both offline and online adjustments (Fig 5C, offline and online). Observe as

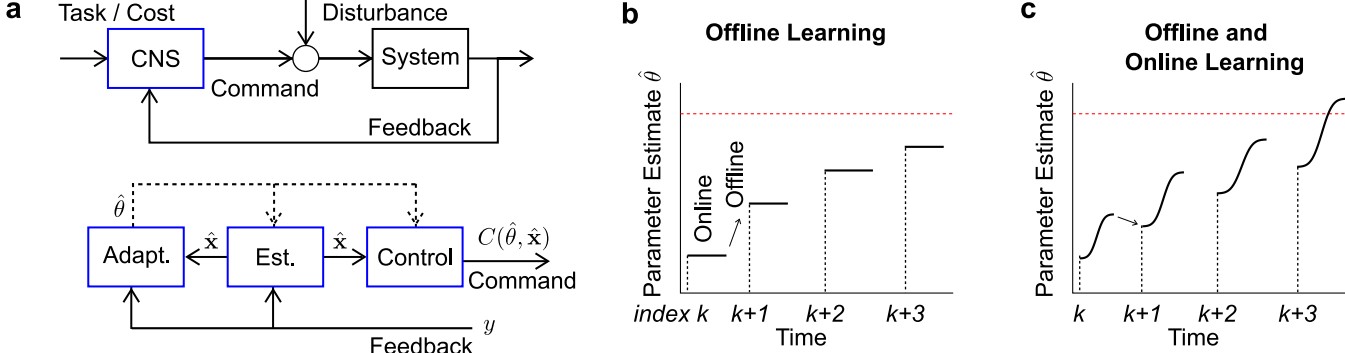

**Fig 5. Schematic Model. a.** Diagram of an adaptive state-feedback control model of the Central Nervous System (CNS). The controller (top, blue) is decomposed into (bottom): a state estimator (Est.), which estimates the current state, an adaptive estimator (Adapt.) which evaluates the model parameters (here the defining coefficient of the force field), and a parametric controller (Control) mapping estimated state into motor commands (see text for details). **b.** Trial-by-trial adaptation model where the estimated parameter changes offline between two trials. Black lines indicate the estimated parameter over time during the trials indexed by *k*. Observe the estimate of the force field parameter is bounded by our experimental results (red line), since anticipatory compensation for the force field was partial. **c.** Trial-by-trial (offline) and online adaptation schemes, here the estimated force field parameter is allowed to change within a trial, possibly beyond the estimate used prior to movement onset.

illustrated schematically in Fig 5C that offline processing in this case could correspond to partial unlearning induced by the random schedule, and that movements could adapt online beyond the value that corresponded to the estimate used during movement planning. We first highlight the presence of two different timescales in participants' data attributable to online and offline mechanisms, and then we develop how this aspect could be explained in the context of adaptive control.

The slow formation of a memory (or prior) across trials, and the fast feedback adaptation component were visible in participants' behaviour by looking at initial angles linked to anticipation, and at the paths length that also encompassed the online feedback correction. Qualitatively, it can be observed in Fig 6A that besides changes in initial angles, later movements tended to exhibit less deviation overall. We took the opposite of the initial angle from CCW trials to map them onto positive values, then averaged aiming angles across force fields and participants to quantify the absolute decay in aiming angles across trials relative to a straight line (Fig 6B). For this variable, the single rate model (Eq 3) was selected according to the BIC cost (blue fit: $BIC_{DUAL}-BIC_{SINGLE}>0$). The dual rate model (Eq 4) applied to the initial angle produced a value of $\beta_2$ that was not significantly different from 0. In contrast, the dual rate model was retained when we fitted the path length (Fig 6C, $BIC_{DUAL}-BIC_{SINGLE}<0$). The two time-constants were significant and different from each other (fast constant -0.83, 95%CI [-1.58, -0.08], slow constant -0.02, 95%CI: [-0.04, -0.0015]).

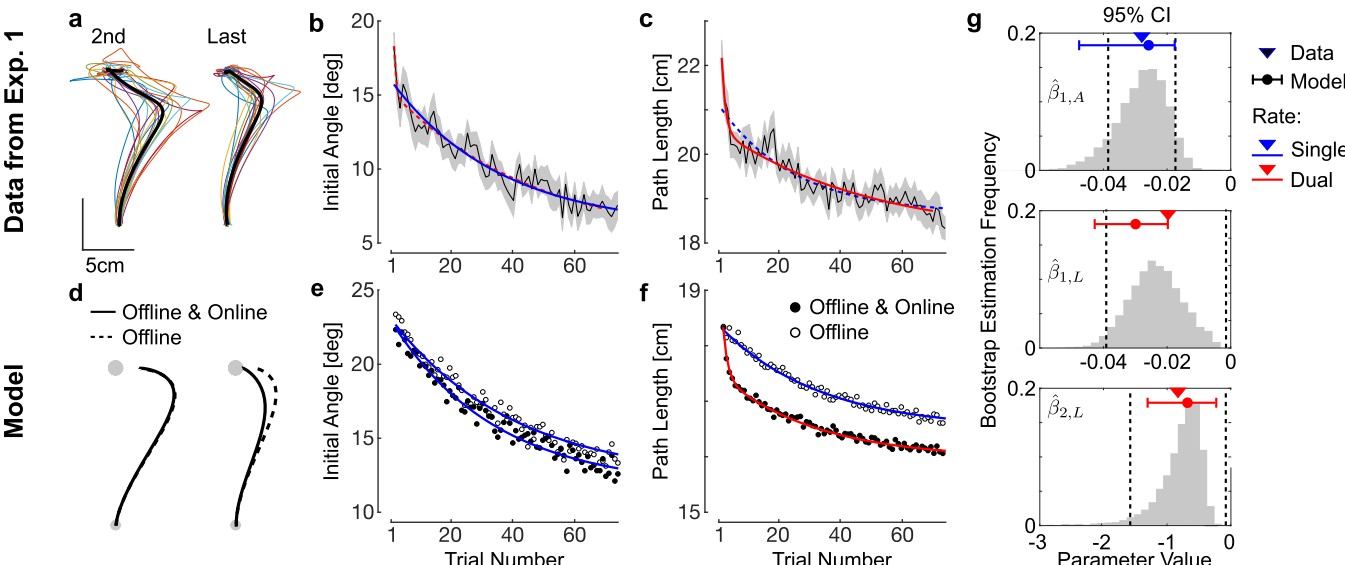

**Fig 6. Comparison between Model and Data. a.** Second and last force field trials from each participant during clockwise perturbations (colour), average trajectory across them (thick black). **b.** Initial angle averaged across participants and SEM. The opposite of initial angles for clockwise perturbations were taken to map onto positive values, and trials were averaged across force fields and participants. **c.** Path length averaged across force fields and participants (mean ±SEM). In panels *b* and *c*, exponential fits with single time constant (single rate, blue) and two time constants (dual rate, red) were fitted, the selected model by BIC criterion is represented in solid trace, the non-selected model is depicted in dashed trace. **d.** Hand traces produced in model simulated with trial-by-trial adaptation only (offline, dashed), and trial-by-trial adaptation plus online adaptation (offline and online, solid). **e.** Initial angle for the two models reproducing exponential decay comparable to the data with single rate exponential fits. **f.** Path length for the two models, with single rate and dual rate exponential models fitted to both models with offline adaptation, or offline and online adaptation, respectively **g.** Empirical and theoretical values of the exponential decay rates. Triangles are the maximum likelihood estimates from the data, dashed vertical lines are the 95% confidence interval estimated from the inverse of the information matrix following standard procedures, histograms are bootstrap estimates obtained by random re-sampling of 18 participants with replacement. Red and blue dots are the parameters obtained based on the simulated angle and path length shown in panels *e* and *f* with the corresponding number of decay rates. Whiskers represent the maximum and minimum values of these parameters after doubling or halving each rate parameter (see Methods for details). Top: single decay rate of initial angle ($\hat{\beta}_{1,A}$); Middle: slow decay rate of path length ($\hat{\beta}_{1,L}$); Bottom: fast decay rate of path length ($\hat{\beta}_{2,L}$).

Fig 6G represents the confidence intervals for each time constant from the selected statistical models of exponential decay across trials (dashed vertical lines), as well as bootstrap estimates obtained by resampling participants with replacement 10,000 times. It should be observed that the slow time constants across initial angles and path length were clearly comparable, and likely related to the same underlying trial-by-trial adaptation. In contrast, the fast time constant of path length decay was not observed in the initial angle, suggesting the presence of a different mechanism. Because initial angles and path lengths were extracted from the same trials, the possibility that they involve different mechanisms is consistent with an online feedback adaptation mechanism.

We show that the model with both online and offline learning rates can reproduce the forgoing observation: a single time constant characterizing the exponential decay in initial angle, and two time constants to explain the decay in path length. Fig 6D shows the two models (offline only, dashed) and offline-online (solid). In the offline model, the path length still exhibited substantive deviation after partial reduction in the initial angle, which can be directly linked to the similar exponential decay in initial angle (Fig 6E and 6F, blue fits). Thus, in this model, the anticipatory components extracted from the initial angle, and the path length evolved in parallel because they both related to offline adjustments. In contrast, with the online learning rate, the feedback correction improved in addition to the offline component. The result was that the path length exhibited two time constants as in the data (Fig 6F, red fit): a slow time constant that again parallels the slow decay in initial angle, plus a faster decay reflecting within-trial improvements in feedback control. The single and dual rate exponential statistical models were fitted to the time series of the control model to verify that similar time constants could be obtained in simulations. The dual rate model fitted to the time series of initial angles did not produce two different time constants with non-overlapping confidence interval, which confirms the observation that the decay in this variable followed a single time constant. Fig 6G shows these time constants in line with participants' data, as well as the range of values obtained for each time-constant after doubling and halving all parameters related to the learning rates (blue and red dots/whiskers: min and max values across all simulations for single rate and dual rate models, respectively). To conclude, we leveraged the paradigm to highlight different timescales in the evolution of kinematic parameters, with a slow evolution for the initial angle, and a dual rate decay for the path length including a comparable slow time constant as well as a faster decay potentially resulting from adjustments of control within trials. This interpretation is supported by a computational model of adaptive control.

## Discussion

Our main findings were that background forces of different directions enabled dual motor adaptation to opposite velocity dependent force fields. However, we uncovered that this mechanism did not directly generalise to different magnitudes. This finding was unexpected: indeed, we initially assumed that a force proportional and in the same direction as the force field should be readily available to anticipate the field magnitude. Contrary to this intuition we found a robust and reproducible interference, even though the formation of separate (but inaccurate) motor memories could be established based on statistical modelling. Hence, the case of force magnitude for short term adaptation is not closed, but these findings clearly open perspectives as discussed below. Finally, we used the data of Experiment 1 (without interference) to highlight that this protocol could be leveraged to identify different timescales of motor adaptation consistent with online adaptive control.

Previous studies have investigated the conditions under which independent cues supported the formation of distinct movement goals [18]. In that regard, one may object that pre-

activation of some muscles by the background forces was not an independent contextual cue, if the same muscles were recruited to counter the force field. We believe that our data resolve this concern: the reason why background forces could be seen as an independent contextual cue is that we were able to evoke adaptation in the positive and negative contexts of Experiment 1, hence the relationship between the cue and the force field was statistically orthogonal by design. Moreover, the interference observed in Experiments 2 and 3 (unexpectedly) pleaded against a simple activation-related anticipation of the force field, which, with different magnitudes, should pose no difficulty. In fact, our data demonstrate the independence between force cues and movement execution, while showing that an association could be rapidly learned in the case of opposite directions.

Our conclusions are partly compatible with the COIN model (*contextual inference*, [36]) but also highlight a limitation of this model. In principle, the COIN model posits that the brain performs Bayesian inference about the context, and uses a weighted average of the different context states to form a motor output. The interference observed with the force magnitude is a counter example to this theory: there was no ambiguity about the contextual cues delivered with the background forces (other than for the catch trial), and the association between the context state (i.e. the magnitude of the background force) and the magnitude of the force field could be made directly. Thus, the tasks of Experiments 2 and 3 are an exception to the set of sensory cues that can be used in the brain for contextual inference.

The first intuition to explain the interference was the possibility that participants used a single representation, which could be close to an average of two or more force fields [3,30,37,38]. However, this explanation would leave out two consistent features of our results: first there was a reliable interaction term in the statistical model of the initial angles across trials, whereas a single internal representation would produce parallel changes for both heavy and light trials without statistical interaction. Second, the initial angle during catch trials suggested that in absolute, the internal estimate associated with the heavy cues was in fact lesser than the estimation associated with light ones. It is conceivable that participants relied on a two-stage strategy: a transient phase that consisted in zeroing the largest motor errors in a non-specific way, such as an up-regulation of feedback gains [22,39], explicit strategies [40], and robust control [41], followed by a second phase with slower dissociation of motor memories associated with the different background magnitudes.

Understanding how humans can learn multiple tasks at the same time has been an attractive question, motivated by the fact that outside of laboratory conditions, we can clearly learn multiple tasks without interference. Indeed, we do not have to unlearn to ride a bike when we walk, and one does not unlearn to play the piano when playing tennis. However, learning to handle clockwise and counter-clockwise force fields at the same time is intriguingly difficult, even with clear cues about the force field direction [5]. Learning opposite velocity-dependent force fields is possible when practiced in different parts of the work space [42]. For similar kinematics, it was shown that different planning conditions were necessary to build different motor memories, as induced by different visual targets following opposite visuomotor remapping, different limb state, or different instructions about movement sequences [10,15–17]. Interestingly, it is not even necessary that a prior hand displacement occurs since the production of a force onto a static handle was sufficient to evoke the formation of distinct motor memories (see [16], Exp. 4). In the context of visuomotor adaptation, the use of different effectors also supports context-dependent learning of opposite perturbations, and limit interference [9].

A common aspect of all previous studies was that the planning conditions associated with different force fields differed explicitly. Yet this aspect was critical, and fundamentally differed from our contribution: here the kinematic plan was similar in all cases and the planning

condition differed only by the counteraction of externally applied loads. It is also possible that the difference across conditions was not confined to pre-movement phase, as participants knew that they had to control their movement and hold their hand still after movement end against a constant load, which may be used as a follow through action enabling separation of motor plans. Taken in the context of current models of sensorimotor adaptation, our results suggest that the neural activity evoked by background forces allowed different planning states for similar kinematic plans and limb configuration without any explicit instruction about prior or follow through actions.

The underlying neural mechanism to expect is not straightforward. The reasoning behind searching for conditions allowing different planning states, in an abstract sense, must be linked to different neural states, but where and how this can occur is not clear. For instance, compound movements do not seem to evoke different preparatory activities in primary and premotor cortex in comparison to the activity evoked when the movement components are performed separately [43], thus putative differences in planning states for movement sequences may originate from other structures. With respect to the Experiments 2 and 3, it is known that activities of single cells in Primary Motor Cortex scale with the magnitude of externally applied loads [44], thus such neural correlate of background force in M1 may not be readily available to separate motor memories during short term adaptation. Hence, the interference mechanism may set constraints on the candidate structures underlying separation of motor memories when the cue was related to the magnitude. Other neural structures that exhibit differences during the planning stage evoked by a background force are candidates for providing input to the temporal evolution of neural activities associated with reach control [45,46].

Our data also set constraints on candidate models of sensorimotor adaptation, and we illustrated that considering offline and online adaptation schemes could explain participant's behaviour [33,35]. It must be emphasized that the fast and slow time constants that we measured in the decay of path lengths do not directly follow from previous models [47,48]. Indeed, in the latter references, the many different timescales still characterized trial-to-trial adaptation, thus these models could not explain differences between early and late stages of the same movement because the fast time scales were always assumed to be slower than the movement time. In our developments, the discrete time series of trial-by-trial estimate (i.e. $\hat{\theta}_k$) could be related to previously suggested models of trial-by-trial adaptation [47–49]. However, by adding the possibility that the force field estimate varied in flight ($\hat{\theta}_k + \Delta\hat{\theta}_t$), our model also explains why initial angles and path lengths of the same movements exhibited different evolutions. It remains possible that peripheral factors or biomechanical interactions produced different decay rates, and we are not in position to reject such possibility firmly. For instance, we cannot reject the possibility that participants used co-contraction in some phase of the experiment. We believe however that the impact of such strategy should be visible in the initial angle as it should be active at movement onset, which was not observed. In any case, a fuller quantification of the impact of feedback adaptation must still be established.

There remain interrogations about the nature of the separation of motor memories enabled by the force cues. Clearly, the presence of interference in Experiments 2 and 3 raises the question of whether and under which condition a proportional relationship can be learned. Likewise, although the separation was clear in Experiment 1, the initial angle after 75 trials showed that adaptation was incomplete, and it would be interesting to study whether straighter trajectories can be recovered after longer training. Our results open perspectives for learning or relearning strategies following neurological deficits: a simple transfer to motor recovery protocols could be by leveraging the impact of gravitational loads known to influence movement planning and control [50]. Indeed, changing limb configuration relative to gravity is a very

simply way to pre-activate different muscle groups, thereby creating "natural" background loads that can be easily manipulated according to intended movements. If gravity-related background forces can support multiple motor memories as in our experiment, such a manipulation is potentially an easy and effective mean to recover flexible control of reaching movements.

## Methods

### Ethics statement

All experimental procedures were approved by the Ethics Board of the host institution (*Comité d'Éthique Hospitalo-Facultaire*, UCLouvain, Belgium). All participants had the age of legal majority and provided written informed consent.

### Experimental procedures

A total of 47 healthy adults took part in the experiments (ages between 23yrs and 41yrs, 21 females), 18 were involved in Experiment 1, 14 in Experiment 2, and 15 in Experiment 3. The following procedure was common across all experiments. Participants grabbed the handle of a robotic device (KINARM, Kingston, Ontario) and were instructed to place a hand-aligned cursor in a start target located in front of them. After a random delay between 2s and 4s, the goal target located 15cm ahead of the start target was filled in red, giving the signal to go. From this time, they had between 600ms and 800ms to reach and stabilise in the goal for at least 1s. Feedback about performance was given by a color code (open red: too fast; filled red: too slow; greed: success), and a counter projected on the screen was incremented in the case of a successful trial. At trial onset and prior to stabilisation in the start target, a constant lateral background force was applied with a linear ramp up of 500ms, and applied throughout the whole trial ($F_x = \phi$). This background load was applied independently of the cursor location, thus participants first had to acquire the start target, or return to it, against the background load. The random delay comprised between 2 and 4 seconds was counted only if the cursor was stable in the start target, meaning that participants awaited the go cue while constantly compensating for the background load. The background force ramped down linearly during 500ms in the end after the trial outcome was notified. During reach movements towards the final target, force field perturbations were added to the background force. The force fields were lateral forces proportional to forward velocity ($F_x = \theta \dot{y}$, see Fig 1A for the reference frame). Visual feedback about the hand-aligned cursor was always presented and direct vision of participants' arm was blocked.

**Experiment 1.** The first experiment tested motor adaptation to clockwise ($\theta > 0$) and counter-clockwise ($\theta < 0$) force fields randomly applied across trials in two conditions of background force illustrated in Fig 1B and 1C: the positive context, in which background force and force field have the same sign ($\theta\phi > 0$), and the negative context in which the background force and the force field have opposite signs ($\theta\phi < 0$). Participants performed three blocks of 60 trials in one context (positive or negative), composed of 50 trials with background and force field, and 5 catch trials per direction of background force without force field, making four different trial types per context: positive and negative values of the background force with their corresponding force field direction, and catch trials with positive and negative values of the background force. Blocks were separated by short pauses on demand to avoid fatigue. After the third block, participants performed a washout block composed of 30 trials identical to catch trials including a background force of random direction (left or right) but no force field (15 per direction), prior to performing a second series of three blocks in the other context (negative or positive, respectively). Switching context was critical to ensure that the force cue could

be considered independent of the force field direction. The order in which participants performed the task in each context was counterbalanced. The parameter values were $|\phi| = 4$N, and $|\phi| = 13$Nsm$^{-1}$.

These blocks were followed by a final set for control analyses, in which clockwise and counter clockwise force field trials were randomly interleaved without any background force. This block was designed to compare initial angles across conditions including or not context-dependent background forces. Participants performed 30 trials, 15 per force field direction, randomly interleaved. This block was always performed in the end.

**Experiment 2.** Building on the results of the first experiment showing flexible association of background force and force field direction, we investigated whether different force field magnitudes could be extracted from background forces also differing by their magnitudes. In this experiment, the initial background force was either light or heavy ($\phi = 2.5$N or $\phi = 5$N, respectively), and the corresponding force field was also light or heavy ($\theta = 7.5$Nsm$^{-1}$ or $\theta = 15$Nsm$^{-1}$, respectively). As in Experiment 1, catch trials with background force but no force field were randomly interleaved. Participants performed six blocks of 60 trials composed of 50 trials with light/heavy background and light/heavy force field randomly interleaved (25 per magnitude), and 5 catch trials per light/heavy background force magnitude but without force field.

**Experiment 3.** This experiment was designed to reveal the interference between motor representations in Experiment 2 by introducing the two force fields sequentially. Participant performed one block with the heavy background and force field trials (50 force field, 10 catch trials), then they performed two blocks identical to Experiment 2, with the light and heavy trials randomly interleaved.

## Data collection and analysis

The KINARM setup allowed sampling of position and force signals at 1kHz. The force applied by participants on the handle reflect the interaction between commanded perturbations and participants' hand, thus under static conditions the sign of the measured force is opposite to the sign of the commanded force. We have kept this convention throughout, such that rightward and leftward background forces were associated to negative and positive force offsets, respectively (see e.g. Fig 1B and 1C). All signals were digitally low pass filtered with dual pass, 4$^{th}$ order Butterworth filter with cut-off frequency set to 50Hz. Velocity was obtained by numerical differentiation of position signals with 4$^{th}$ order centered, finite difference algorithm. Traces were aligned to the moment when the hand-aligned cursor exited the start target defined as the reach onset.

Two parameters were extracted from hand kinematics: the initial angle measured as the angle between the $y$−axis and the line joining the start target and the cursor at the crossing of a virtual threshold corresponding to one third of the distance between the start and goal targets. Our previous reports documented that changes in EMG leading to improvements in feedback corrections occurred after ~250ms following reach onset [34]. This moment typically occurs after the crossing of the virtual threshold, such that the initial angle can be linked to anticipatory compensation for the force field measured prior to the effect of feedback adaptation. The second extracted parameter was the path length calculated as the integral of the norm of the velocity vector. Traces were truncated at 600ms following reach onset. Observe on continuous average traces (Figs 1B and 1C and 2B) that trials were typically completed at that time. The interquartile range of absolute forward velocity at that time across all participants from Experiment 1 was [0.01, 0.044]m/s.

The main statistical comparisons were performed on the initial angles based on paired t-tests. For these comparisons, we estimated the effect sizes as the mean difference across

conditions divided by the standard deviation of the pooled data [51]. Calling $d$ the vector of differences for each participant, $\bar{d}$ its average, $\sigma$ the standard deviation of the difference, and $n$ the sample size, the power was estimated as

$$P(\text{reject } H_0 | H_1 \text{ true}) = 1 - \Phi\left(T - \frac{\bar{d}}{\sigma/\sqrt{n}}\right), \tag{1}$$

with $T$ the percentile $1-\alpha$ of the zero-mean, unit variance Gaussian cumulative density function $\Phi(.)$. Statistical significance and power calculations were based on the level $\alpha = 0.005$.

Experiments 2 and 3 revealed subtler effects for which we used mixed linear models to include all individual trials and increase statistical power. This is possible by considering models with a random intercept to account for idiosyncrasy [52]. The general form of mixed linear models was as follows:

$$Y_{i,j} = Xb + s_j + \varepsilon_{i,j}, \tag{2}$$

with $Y_{i,j}$ representing the dependent variable from trial $i$ of participant $j$, $X$ being the fixed predictor matrix, $b$ the coefficients of fixed predictors in the linear model, $s_j$ were the fitted random offsets associated with each participant, and $\varepsilon_{i,j}$ was the model residual associated with each trial and participant. For the data of Experiment 2, the matrix of fixed predictor consisted of an intercept, a categorical variable corresponding to the magnitude (heavy or light), the trial index, and an interaction term between the trial index and the magnitude. This model was fitted to the initial angle and to the forward velocity measured at the same time. Mixed models were also used to compare the results of Experiments 2 and 3, with a predictor matrix composed of an intercept, two categorical variables for the force magnitude and the experiment, and an interaction between them. This model was fitted on the initial angle during catch trials.

Statistical models of the exponential decays in initial angle and path length from the first experiment were obtained by maximum likelihood estimation of the parameters in the following exponential models. The single rate model was as follows:

$$f(x) = \alpha_0 + \alpha_1 e^{-\beta_1 x} \tag{3}$$

where $f(x)$ was the dependent variable (initial angle or path length), and $x$ was the predictor (trial index). The dual rate model was as follows:

$$f(x) = \alpha_0 + \alpha_1 e^{-\beta_1 x} + \alpha_2 e^{-\beta_2 x} \tag{4}$$

We fitted the two models and assessed them based on Bayesian Information Criterion (BIC) as model selection technique. We observed that when Eq 4 was fitted to data with only one time constant, either the second exponential decay parameter was found not-significant, or it was equal to the first one leading to similar variance accounted for as from Eq 3, with an extra parameter typically rejected by the BIC. The BIC was estimated as follows [53]: defining the residuals as $r_i$, $i = 1,\ldots,N$, and $k$ the number of parameters in the model (3 or 5 for Eqs 3 or 4, respectively), the approximation was:

$$BIC = N \ln\left(\frac{\sum_{i=1}^{N} r_i}{N}\right) + k \ln N. \tag{5}$$

Since Eq 5 is a *cost*, the best model is the one with the smallest BIC.

Trust regions and 95% confidence intervals for the model parameters were obtained with two different techniques: the first was based on standard estimation of the covariance matrix of the fitted parameters based on the Gaussian approximation of the posterior distribution of

the parameters [54]. The second was by bootstrapping: we randomly selected 18 participants from the initial population with replacement such that some participants were picked multiple times and others were left out at each iteration. The resampled population was then used to calculate the same parameters and build a histogram of the distribution of the parameters across iteration [55]. The procedure was repeated 10,000 times. There were iterations for which the model fit did not converge to a reliable estimate. These missing values were simply discarded, and it occurred in only 1.57% of cases. As a result, the distribution of parameter estimates from this bootstrap procedure included 9,843 values. These distributions were compared with the frequentist confidence intervals.

## Computational adaptation model

We simulated the same adaptive control model published in [33], and only manipulated the learning rate to reproduce the presence of a single rate exponential decay in the initial angle, and of dual rate exponential decay in the path length. This presence of multiple timescales for kinematic variables of the same movement suggested that distinct mechanisms could underlie slow trial-by-trial adaptation, and fast within trial adjustments mediated by feedback adaptation. We show that the adaptive control model with online estimation can indeed accommodate these experimental observations.

The biomechanical model and adaptive controller were described in detail in [33], and reused in this study. Briefly, the model is composed of a state-feedback controller derived in the context of Linear Quadratic Gaussian (LQG) given the model parameters, which were updated during movement. We define the state vector as $\mathbf{x}_t$, including two-dimensional position, velocity and commanded force. The uncertain parameter from the perspective of the controller is the force field magnitude, $\theta$. The controller is thus a parametric function of the estimated state $\hat{\mathbf{x}}$, and estimated force field parameter value $\hat{\theta}$, defined as $C(\hat{\theta}, \hat{\mathbf{x}})$. The adaptive control model proceeds as follows (Fig 5A): sensory feedback is used to estimate the state given the current estimate of the parameter (Kalman filter), which produces an error signal that is used in turn to update the parameter estimate. A new controller is derived at the next time step, including the novel parameter value.

We define two indices linked to time: $k$ refers to the trial index and therefore corresponds to a discrete time series of trial-by-trial changes in parameters, while $t$ refers to continuously changing parameters over time during movement execution. We consider two adaptation models. The first one uses an estimate $\hat{\theta}_k$ for trial $k$ that is fixed within the trial, and updates the state-feedback controller across trials (Fig 5B, trial-by-trial adaptation). For this model, the control and adaptation processes can be described as:

**Model 1.**

$$\text{Control (online)} : \qquad u_t = C_k(\hat{\theta}_k, \hat{\mathbf{x}}_t), \qquad (6)$$

$$\text{Adaptation (offline)} : \quad \hat{\theta}_{k+1} \leftarrow f(\hat{\theta}_k, \mathbf{e}). \qquad (7)$$

Eq 7 expresses that the parameter update for the $k+1^{\text{th}}$ trial is a function of the previous estimate and of some error metric (**e**) from the previous trial. The second adaptation model includes an online update of the controller based on time-varying changes in the model parameter:

**Model 2.**

$$\text{Control (online)} : \qquad u_t = C_t(\hat{\theta}_t, \hat{\mathbf{x}}_t), \qquad (8)$$

$$\text{Adaptation (online)}: \quad \hat{\theta}_{t+1} \leftarrow f(\hat{\theta}_t, \mathbf{e}), \tag{9}$$

where $\hat{\theta}_t = \hat{\theta}_k + \Delta\hat{\theta}_t$, the latter term capturing within-trial adjustments, and $C_t(\hat{\theta}_t, \hat{\mathbf{x}}_t)$ expresses that the controller is updated over time (Fig 5C, both online and offline). Observe that the indices of the controllers in Eqs (6) and (7) express that one is fixed during a trial, and therefore has the same index as the trial, $k$, whereas the other is time varying including within a trial, and as a consequence has an index that refers to continuous time ($t$).

For the two models, the learning rule was the same as the one described in [33]. Let $\mathbf{e}_{t+1}$ designate the one step prediction error under the current estimate of $\theta$, defined as the difference between the predicted state $\hat{\mathbf{x}}_{t+1}$ and the measured state through sensory feedback, $\mathbf{x}_{t+1}$. The update obtained from least-square identification was [56]:

$$\hat{\theta}_{t+1} = \hat{\theta}_t + \gamma_i \frac{\partial \hat{\mathbf{x}}_{t+1}}{\partial \theta} \mathbf{e}_{t+1}. \tag{10}$$

Eq 10 is the learning rule describing how the measured error is mapped onto parameter updates. Both models were simulated with the same discrete time series of trial-by-trial learning determined by $\hat{\theta}_k$, $k = 1,2,\ldots,N$. This discrete time series was derived with $\gamma_1$ updating the parameter online, but not the controller, and using the value of the last time step ($T$) of the previous trial as $\hat{\theta}_{k+1} = \hat{\theta}_k + \Delta\hat{\theta}_T$. Thus $\gamma_1$ represents the learning rate that explains trial-by-trial updates. In addition, in Model 2, the estimate started from $\hat{\theta}_k$ at the first step for each trial, and was updated during movement as well as the controller with an online learning rate $\gamma_2$. Hence this parameter captures the changes occurring within a trial in addition to the inter-trial rate $\gamma_1$.

We did not perform a systematic sweeping through different values of $\gamma_1$ and $\gamma_2$ to fit the data quantitatively, the reason being that several simplifying assumptions can clearly produce differences between simulations and data that are not worth explaining in detail. Instead, we concentrated on the key feature of the data, namely single rate decay of aiming angle and dual rate decay of path length, and reproduced it while taking the empirical observations as constraints: possible values of $\gamma_1$ were bounded by the observed initial angle at the end of 75 trials. Moreover, participants' data exhibited a significant asymptote of ~6deg. This could be accommodated by a decreasing value of $\gamma_1$. In practice, the numerical values were from 0.001 to 0. In contrast, $\gamma_2$ must increase across trials to produce improvements in feedback adaptation [33], which was implemented between the values of 0 and 0.2. The offline and online learning rates were varied between their minimum and maximum values across trials with first order dynamics with time constants 0.02 and 0.4. These values were selected as they produced consistent fit to human data. Finally, we characterized the sensitivity of the model predictions to the ranges of $\gamma_1$ and $\gamma_2$ and their rates by doubling and halving all numerical values of their domain, to show that the model properties were qualitatively similar across a broad set of values, and that they were not a fragile feature of the model.

The data was deposited in DRYAD [57].

## Dryad DOI

https://doi.org/10.5061/dryad.b2rbnzsk3 [57]

## Author Contributions

**Conceptualization:** Frédéric Crevecoeur, James Mathew, Philippe Lefèvre.

**Data curation:** Frédéric Crevecoeur, James Mathew.

**Formal analysis:** Frédéric Crevecoeur.

**Funding acquisition:** Frédéric Crevecoeur, Philippe Lefèvre.

**Investigation:** Frédéric Crevecoeur, James Mathew, Philippe Lefèvre.

**Methodology:** Frédéric Crevecoeur, Philippe Lefèvre.

**Project administration:** Frédéric Crevecoeur, Philippe Lefèvre.

**Resources:** Frédéric Crevecoeur, Philippe Lefèvre.

**Software:** Frédéric Crevecoeur.

**Supervision:** Frédéric Crevecoeur, James Mathew.

**Validation:** Frédéric Crevecoeur, James Mathew, Philippe Lefèvre.

**Visualization:** Frédéric Crevecoeur.

**Writing – original draft:** Frédéric Crevecoeur.

**Writing – review & editing:** Frédéric Crevecoeur, James Mathew, Philippe Lefèvre.

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
