## [Editor Report · Decision Letter 0]

9 Mar 2022

Dear Dr. Crevecoeur,

Thank you very much for submitting your manuscript "Separability of Human Motor Memories during Reaching Adaptation with Force Cues" (PCOMPBIOL-D-22-00287) for consideration at PLOS Computational Biology. We apologize for the delay in responding.

As with all papers, your manuscript was reviewed by members of the editorial board. Based on our initial assessment, we regret that we will not be pursuing this manuscript for publication at PLOS Computational Biology. The experimental results provide new insights into sensorimotor adaptation dependent on context, and model seems appropriate. Nevertheless in its present form the model, already presented in your previous Crevecoeur etal eNeuro 2020, does not necessarily provide a novel advance vis-a-vis the computational analysis of a biological phenomena. 

We are sorry that we cannot be more positive on this occasion. We very much appreciate your wish to present your work in one of PLOS's Open Access publications. Thank you for your support, and we hope that you will consider PLOS Computational Biology for other submissions in the future.

Sincerely,

Lyle Graham

Deputy Editor

PLOS Computational Biology

---

## [Decision Letter · Decision Letter 1]

6 Jul 2022

Dear Dr. Crevecoeur,

Thank you very much for submitting your manuscript "Separability of Human Motor Memories during Reaching Adaptation with Force Cues" for consideration at PLOS Computational Biology.

As with all papers reviewed by the journal, your manuscript was reviewed by members of the editorial board and by several independent reviewers. The reviewers were generally positive about the work, but highlighted a number of issues that need to be addressed.  In light of the reviews (below this email), we would like to invite the resubmission of a significantly-revised version that takes into account the reviewers' comments.

We cannot make any decision about publication until we have seen the revised manuscript and your response to the reviewers' comments. Your revised manuscript is also likely to be sent to reviewers for further evaluation.

Sincerely,

Adrian M Haith

Associate Editor

PLOS Computational Biology

Lyle Graham

Deputy Editor

PLOS Computational Biology

Reviewer's Responses to Questions

**Comments to the Authors:**

Reviewer #1: This work investigates whether different background forces can, as cues, allow for separate, simultaneous adaptation to different force-fields applied during movement. In a set of experiments, this work finds evidence that first, oppositely-directed background forces can indeed cue opposing force-fields and, to a lesser extent, samely-directed background forces of different magnitudes can cue samely-directed force-fields of different magnitude. The authors use these data to demonstrate a model of adaptation with two timescales: one timescale representing online adaptation (within each movement) and another representing feedforward adaptation (from one trial to the next). The use of background forces as cues for different motor memories is a very interesting topic, the experiments are well-designed, the paper clear, the figures crisp. I have a number of questions regarding the framing/ interpretation of the results, and how the computational model relates to the experiment at hand.

Framing: As the manuscript discusses, previous work has shown that different force-fields can be cued by different prior or follow-up movements (e.g. refs 16,17). Here, the background forces are imposed before and remain on after the movement is executed. Could the present findings be understood within the concept of this previous work (or, maybe better, expanding the concept)? The common thread being, there is a motor plan before and after the movement that can act as a cue; whether this plan is to move from/to a different direction or to hold still against a different background load?

Lines 125-129 (and Figure 1d): because absolute angles are analyzed here, a smaller absolute angle could also be due to increased co-contraction (because of the added background load). How can we distinguish effects of actually learning the corresponding opposing fields vs. having smaller errors because of increased stiffness? (for this reason, I find the catch-trial findings eliciting oppositely-directed errors more convincing).

In similar lines: how can we distinguish between the part of the learning curves is due to stiffening vs. learning each force-field? In particular, the rapid component of the learning curve for path length (Figure 6c) could represent stiffening (especially later in movement) upon the introduction of the force-field. I suggest a brief discussion of these issues.

Figure 1 can benefit from the addition of a schematic illustrating the different phases of the experiment. While details can be inferred by a careful reading of the methods, a schematic could make this much easier.

Was the washout block (described in lines 428-431) analyzed? It would be particularly interesting to see if the different background forces during that period elicit performance reflecting the learning of one force-field or the other.

Testing both conditions in all individuals is very reasonable. Were there any order effects however?

Modeling: I don’t totally get how the two parts of this paper – different force-fields cued by different background loads on one hand, vs. a (very interesting) model encompassing both an online and offline component – blend together. The paper mentions that the data in Experiment 1 give a “novel opportunity to illustrate the model”. Can the authors elaborate on this? The model has been shown and tested in dedicated experiments in the authors’ recent work (ref. 33) – what regarding the present experiment design allows us to understand the model better? Is there something regarding background force cues that makes this dataset better suited to demonstrate the model than a garden-variety force-field adaptation task? From the other side, are there specific insights about learning multiple force-fields at the same time using force cues that are demonstrated through the model?

Minor comments/questions

Abstract, Lines 25-28. This makes it sound like there was no interference in the opposite-direction case – however, to make that claim one needs to compare how quickly the two opposite force-fields are learned separately – instead, it might be better to say “increased interference”.

Not an important issue, but I was thinking the word “offline” to represent trial-to-trial adaptation might be confusing to some readers, as it may cue them to think of offline gains in learning/consolidation.

The findings in Experiment 2 are definitely interesting – my first thought was that participants learn the average force-field, but the discussion addressed this well. Could participants be learning one field but scale their response by the force cue? Figure 2c suggests that the adaptation to the stronger force-field is privileged over adaptation to the smaller one – smaller errors – why would that be?

Lines 472-473: is it necessary to truncate traces at 600ms to calculate path length? What fraction of trials was completed by that time?

Line 586: which data were used for that purpose?

Reviewer #2: This study investigates whether distinct background forces are sufficient cues to allow for dual adaptation to opposing forces fields. The authors found partial dual adaptation, errors reduce across training and expected catch trials are produced, when the background forces are in opposing directions although this is less the case when the background forces are in the same direction but different magnitudes, likely due to interference. The papers also demonstrates that the time courses of dual adaptation to FF with opposing backgrounds appears to be made of a combination of offline and online adaptation. The results are very interesting and a nice contribution to our understanding of the contextual cues that brain needs to concurrently create distinct motor memories. I particularly like Experiment 3; which is a somewhat novel way (as far as I know) of exploring interference and dual adaptation.

I was informed that this is a revision, but I wasn’t one of the original reviewers and didn’t see the reviewers or replies. But I don’t have much to add but only a few comments.

The figures are great, and I appreciate the individual data for the catch-trials. But I think the paper would benefit from seeing individual data for training phases, especially for Experiment 2, where the effect could be driven by a subset of participants especially since the sample size is bordering on too small. I know individual curves may be too messy, so maybe fitting an exponent decay with an asymptote and plot the rate of change and asymptote values. Such fits are more sensitive for looking at differences in in learning rates (across groups, as a function of interference etc) than just averaging the first block of trials (which these authors don’t do here but is fairly common).

I can imagine that how early the background force is experienced prior to movement onset could make a difference to how useful the cue is, especially when differentiating between a small and large background forces, although this is not mentioned. In fact, I had no idea when or what event triggered the background force until I read the methods at the end and I’m still not 100% sure. The methods indicate the background force was applied (across 500 ms ramp) prior to target onset but before participants stabilized their hand onto the home to start the movements. What does stabilisation entail? My understanding is that people acquire the start target, and that then the background force ramps up but is there an additional requirement that they had to keep their hand within a window of the start position (make sense) which would have been a “challenge” since they have to overcome the background force. How long did it take for participants to stabilize (given that the background force should have made it more challenging)? And likewise, what was the average amount of time people experience the full force of the background prior to movement onset. Once the background force ramped up, how long until the target appears. Did the amount of time that background force was experience differ for the different magnitudes of forces given that it would been easier to stabilise the hand at the start position for the light background force than the heavy one.

It may be useful to describe background force timing a bit earlier than the methods or make use of Fig 1 and its legend to provide a bit information about this timing relative to target onset and start stabilization. The timecourse for b and c looks more like a schematic, so not sure whether to rely on it to indicate that actual time that the full background force was applied prior to movement onset (0).

Line 385, the term “interrogations” seems odd. I understand what you mean but its not match the context and has a negative connotation.

Line 26. The “t” in their is missing

**Have the authors made all data and (if applicable) computational code underlying the findings in their manuscript fully available?**

Reviewer #1: Yes

Reviewer #2: **No: **Not yet -they will wait until publication. But ideally shouldn't be necessary to wait until publication.

PLOS authors have the option to publish the peer review history of their article (what does this mean?). If published, this will include your full peer review and any attached files.

Reviewer #1: No

Reviewer #2: **Yes: **Denise
---

## [Decision Letter · Decision Letter 2]

30 Sep 2022

Dear Dr. Crevecoeur,

We are pleased to inform you that your manuscript 'Separability of Human Motor Memories during Reaching Adaptation with Force Cues' has been provisionally accepted for publication in PLOS Computational Biology.

Best regards,

Adrian M Haith

Academic Editor

PLOS Computational Biology

Lyle Graham

Section Editor

PLOS Computational Biology

Reviewer's Responses to Questions

**Comments to the Authors:**

Reviewer #1: Thank you for your clarifications and adjustments to the paper. I particularly appreciate explaining and motivating how the present data are uniquely suited to illustrate your model. I have no other comments.

Reviewer #2: I'm satisfied with the authors' replies to my concerns.

**Have the authors made all data and (if applicable) computational code underlying the findings in their manuscript fully available?**

Reviewer #1: None

Reviewer #2: Yes

PLOS authors have the option to publish the peer review history of their article (what does this mean?). If published, this will include your full peer review and any attached files.

Reviewer #1: No

Reviewer #2: No

---

## [Editor Report · Acceptance letter]

24 Oct 2022

PCOMPBIOL-D-22-00287R2 

Separability of Human Motor Memories during Reaching Adaptation with Force Cues

Dear Dr Crevecoeur,

I am pleased to inform you that your manuscript has been formally accepted for publication in PLOS Computational Biology. Your manuscript is now with our production department and you will be notified of the publication date in due course.

With kind regards,

Olena Szabo
